# Ubiquitous Electron Transport in Non-Electron Transfer Proteins

**DOI:** 10.3390/life10050072

**Published:** 2020-05-20

**Authors:** Stuart Lindsay

**Affiliations:** Biodesign Institute, Department of Physics and School of Molecular Sciences, Arizona State University, Tempe, AZ 85287, USA; stuart.lindsay@asu.edu

**Keywords:** bioelectronics, protein electron transport, protein electron transfer, electronic properties of proteins, single-molecule electronic devices, molecular electronics

## Abstract

Many proteins that have no known role in electron transfer processes are excellent electronic conductors. This surprising characteristic is not generally evident in bulk aggregates or crystals, or in isolated, solvated peptides, because the outer hydrophilic shell of the protein presents a barrier to charge injection. Ligands that penetrate this barrier make excellent electrical contacts, yielding conductivities on the order of a S/m. The Fermi Energy of metal electrodes is aligned with the energy of internal electronic states of the protein, as evidenced by resonant transmission peaks at about 0.3V on the Normal Hydrogen Electrode scale. This energy is about 0.7 V less than the oxidation potential of aromatic amino acids, indicating a large reduction in electrostatic reorganization energy losses in the interior of the proteins. Consistent with a possible biological role for this conductance, there is a strong dependence on protein conformation. Thus, direct measurement of conductance is a powerful new way to read out protein conformation in real time, opening the way to new types of single molecule sensors and sequencing devices.

## 1. Introduction

It is generally believed that proteins are excellent insulators. Before delving into what is meant by this statement, it is helpful to make a distinction between electron transfer and electron transport (following Ing et al. [1]). Almost all of the important processes in biochemistry involve electron transfer (Figure 1a), in which an electron (or hole) is localized on some site on the protein, as part of the overall electron transfer chemistry of a cell. Electron transport, on the other hand, is the passage of charge through a protein from an external donor to an external acceptor (Figure 1b). It is not usually considered to be a biologically important process. An exception is the remarkable conducting filaments that some deep-soil bacteria produce in order to get rid of electrons (by reducing iron oxide particles in soil). In the case of *Geobacter sulfurreduccans*, 3 nm diameter filaments extend tens to hundreds of microns and have conductivities of around 5 S/m [2,3], comparable to that of a semiconductor like germanium. In the case of cable bacteria, conducting filaments can extend for centimeters [4]. These are rather special cases, and there is no currently known evolutionary pressure for non-electron transfer proteins to form conductive structures in general.

In order to discuss the conductivity of proteins, we first consider the atomic requirements for a classic conductor like a metal or a semiconductor. Figure 1c shows the simplest possible solid-state tight binding model. It consists of an infinite chain of “atoms”, each with an orbital energy $E_{0}$. The energy of the interaction between neighboring atomic sites is given by a quantity τ. This model is readily solved [5] to show that the connected atoms give rise to a band of delocalized electronic states of width, W=2τ. If an electron is injected with an energy ($E_{F}$) that coincides with unoccupied states in this band, the conduction is metallic, with delocalized electrons. The possibility of delocalized conduction bands in proteins was proposed nearly 80 years ago by Szent-Gyorgyi [6], but existence of the required long-lived quantum-coherence in proteins is extremely controversial [7].

In most molecular systems coupled to electrodes, it is believed [8] that the electron energy, $E_{F}$, is such that the electron is not injected into/out of allowed states (i.e., $E_{F}$ is not in the range of $E_{0}\pm W/2$). In that case, transport occurs by tunneling [1,5] with a transmission probability P∝exp(−2κd), where *d* is the distance and $\kappa =\sqrt{\frac{2m(E_{0}-E_{F})}{\hbar^{2}}}$ where m is the mass of an electron, and ℏ is Planck’s constant divided by 2π. $E_{0}-E_{F}$ is typically a few electron volts, so the decay constant, 2κ, is on the order of 1 ${Å}^{-1}$. This means that tunneling is restricted to distances of a few Angstroms, and cannot account for long-range transport (e.g., in bacterial filaments). Thus, we see that long-range transport requires resonant ($E_{0}$ close to $E_{F}$) injection of electrons. Even with resonant injection, there is a further complication for soft, weakly bonded, molecular solids like proteins. Electrostatic reorganization of the environment lowers the Coulomb energy of an electron on a particular site, trapping it (this is the Marcus [9] reorganization energy, λ). In the case where λ≫τ, the broadening of atomic states, W, is no longer dominated by τ, but rather by λ, which couples each atomic site to dissipative environmental fluctuations. This leads to shifts of the unperturbed atomic energy ($E_{0}$) by an amount in the order of λ. An electron, remaining on a site for long enough, loses energy ~λ to the environment, to become trapped on that site. In order to move to an adjacent site, it must overcome this barrier via thermal excitation, with a hopping probability $\exp\left(\frac{-\lambda }{4kT}\right)$ [9]. If τ≪λ, then hopping is the only available transport mechanism. In a polar solvent like water, λ≈ 1 eV, compared to a value of 0.1 eV for 4kT at room temperature. Thus, the probability of hopping is very small, so conventional hopping is also unlikely to account for long range transport.

As we shall see, long-range electron transport is ubiquitous in proteins, despite the theoretical impediments discussed above. New theories are needed.

## 2. Evidence That Proteins Conduct

The laboratory of David Cahen at the Weizmann Institute in Israel has been investigating bioelectronic devices in which layers of protein are sandwiched between electrodes (as shown in Figure 1b), for many years. The group has published extensively [10,11,12] and a summary of one of their surveys is given in Figure 2. This Figure is based on a published scatter plot (Figure 2 of Amdursky et al. [11]), summarizing the measured current density, j, as a function of layer thickness for alkane chains, aromatic polymers (molecular wires) and proteins. I have fitted their data points with exponential decays of the form $J=J_{0}\exp\left(-\beta d\right)$ (β is the equivalent of 2κ in the discussion of tunneling above). On this plot of logj verus *d*, the slope (when converted to natural logarithm units) gives the decay constant, β in units of nm^−1^. The intercept at d=0 is $J_{0}$, a quantity that is proportional to the inverse of the contact resistance. Quite remarkably, at distances >7 nm, proteins (β= 1.9 nm^−1^) outperform molecular wires (β= 4.3 nm^−1^) by a wide margin. Given that, bacterial filaments aside, there is no known biological driving force for long-range transport, it is surprising that proteins outperform the best efforts of chemical synthesis. On the other hand, protein transport is limited by a large contact resistance ($J_{0}$ = 10^−13^ A/nm^2^ for proteins compared to 10^−9^ A/nm^2^ for alkanes and molecular wires). Almost all of these studies were carried out using proteins containing redox centers. However, one experiment that compared transport in holo-azurin (copper containing) with apo-azurin (copper deficient), found that the room-temperature conductance was similar (but the temperature dependence was not) [12].

## 3. Single Molecule Measurements

There have been a number of single molecule measurements of protein conductance [13,14,15,16,17,18,19,20,21,22,23] (and see the previously cited reviews [10,11,12] for other examples), but many of them suffer one or more of the following limitations that we sought to address in our recent studies:(a)Water is often present (and, if not, should be to maintain protein folding), giving rise to the possibility that current is carried by ions. To address this, we carried out measurements with electrodes submerged in electrolyte, and under electrochemical potential control, such that electrode potentials are maintained outside the region where Faradaic currents are generated.(b)Almost all of the proteins in the prior studies referred to were redox active, forming part of the biological electron transfer chain. We were concerned that some (unknown) nanoscale mechanism might involve transport via rapid reduction and oxidation of the redox active sites. For this reason, we chose to study proteins that were electrochemically inert, eliminating the complications of redox co-factor mediated transport.(c)Non-specific adsorption and denaturation of proteins are common problems on electrode surfaces, often overcome by treating the surface with specific ligands for the target protein [24] an approach we have adopted in our work, testing the specificity with non-binding control proteins, in order to show that binding is selective on the electrode surface.(d)The surface chemistry of proteins is complex, so, as the use of specific binding ligands is absent, the chemical nature of the contact between the metal and the protein is unknown.

Using these methods, we found nS conductance over ~10nm distance in a large protein [25] (integrin) when, and only when it was bound to one of the two electrodes by a specific bond (i.e., cyclic RGD peptide). A non-binding mutant integrin gave no signal at all. We went on to carry out a systematic study of six proteins (three types of antibody, a Fab fragment, streptavidin and repeated the measurements on integrin protein) [26]. In addition to using specific ligands for each of the proteins, we studied bare electrodes and electrodes rendered hydrophilic with mercaptoethanol, in order to elucidate the role of non-specific contacts [26]. We made these measurements using an electrochemical scanning tunneling microscope (STM), as shown in Figure 3a. The electrodes were functionalized with the appropriate ligand and maintained under potential control. Figure 3a shows biotin functionalized electrodes, capturing a streptavidin protein. The gap was set to be larger than the ligands but smaller than the protein, so that no current was passed in the absence of the protein. The junction conducts once a protein is trapped between tip and substrate, at which point current-voltage curves were collected every 0.4 s over a range of ±0.2 V. The process was then repeated at many different points on the substrate to collect about 1000 IV curves in a given experiment. Figure 3b shows an example of a typical IV curve. Most (80%) of such curves were reproducible on repeated sweeps (black points are sweeping up, red points are sweeping down) and linear (the green line is a fit yielding a conductance of 0.36 nS in this example). The noise labeled “TN” has been observed in all proteins studied at all gap distances, and it is almost always only observed at biases above 100 mV. TN here stands for “telegraph noise”, because the current, at a constant bias, jumps between two levels, like a telegraph signal [25]. It is associated with contact fluctuations driven by the electric field at the contact point. While telegraph noise is useful as an indicator of a single molecule in the junction, it is the “quiet” region below 100mV that is of most interest, as we will discuss below.

## 4. Contacts and Protein Conductance

Repeated measurements of single-molecule conductance (obtained from the slope of IV curves as in Figure 3b) generate distributions that reflect the range of contact geometries. Examples of such distributions are shown in Figure 4. Proteins connected by one specific contact (for example, an epitope binding an antibody) and one non-specific contact (for example, a hydrophilic residue on the probe in contact with a hydrophilic amino acid on the protein surface) yield a log-normal distribution (Figure 4a), with a peak value between 0.2 and 0.4 nS. This is a surprisingly small range of conductances for a series of very different proteins of different sizes. The antibodies, which are capable of binding at two sites, show a bimodal distribution (Figure 4b), with a second peak at about 10× the conductance of the first. We generated a Fab fragment (half-antibody) from one of the antibodies, switching the distribution from bimodal (Figure 4b) to a single-peak log-normal distribution (Figure 4a). From this, we conclude that the higher conductance peak at 2nS arises from two specific contacts, so that the current path must be into one Fab fragment of the antibody and out of the other. This is a distance of about 13 nm. The cross-sectional area of one Fab arm is about 10 nm^2^ so that a 2 nS conductance corresponds to a conductivity of about 2 S/m (but see below). Remarkably, this is similar to the conductance of bacterial filaments [2] that have evolved to carry charge, and contain closely stacked chains of hemes [3].

## 5. Distance Dependence of Protein Conductance

Held by two specific contact points (right side of Figure 4), the path through the molecule does not change until the gap becomes too large to contact the protein. In consequence, we observed that the conductance of these junctions does not change with the electrode gap size [3], in complete contrast to what is observed for tunnel transport. However, this is not the case when one of the contacts is non-specific and can therefore be made at almost any point on the antibody surface (left side of Figure 4). In consequence, the first (lower conductance) peak in the conductance distribution shows a small, but statistically significant decrease in conductance with distance [27]. (We carried out experiments to characterize the non-specific interaction, so that we can be reasonably sure that the contact resistance does not change with gap size.) Data are shown as the open circles in Figure 5, and the slope is fitted (dotted line) by the very small value of β of 0.095 nm^−1^, about a tenth of the value of β for a molecular wire in the hopping transport regime [28] (triangles, Figure 5). This small value stands in sharp contrast to the value obtained by fitting (solid line) data points (solid circles) for a number of measurements on small peptides [29,30,31,32,33], for which we obtain β= 4.5 nm^−1^ (in these break-junction measurements, peptides are under tension, and so may be extended). How can components of the protein (peptides) be a factor ~exp(45) less conductive per unit length than the intact protein? A hint is given by the different conductances at zero length, G(0). G(0) for the peptides is 230 nS, but 0.46 nS for the proteins. This indicates a substantial contact resistance arising from an insulating barrier at the surface. Note that this is qualitatively consistent with the macroscopic measurements summarized in Figure 2. The conductance, extrapolated to zero gap, of the proteins is similar to the conductance of about 2 nm of hydrated peptide (Figure 5), suggesting that the insulating barrier consists of an outer shell of hydrated amino acids. 

The data that give these tiny values of β can just as easily (and probably more appropriately) be interpreted in terms of a resistance that increases linearly with length. Replotting our data for the antibodies and Fab fragments this way yields a resistance, R, given by R=(0.53+0.34d) GΩ where d is the path length in nm. Using again a 10 nm^2^ cross-section and the differential value of 0.34 GΩ/nm leads to a value of 0.3 S/m; less than the value estimated above, but still surprisingly high. 

## 6. Energy of the States Responsible for Transport

We noticed that the conductance of proteins in tunnel gaps depends strongly on the metal(s) used as contacts. We studied this systematically, using rest-potential measurements to calibrate the electron-injection potentials for a particular combination of metals and surface functionalizations. We also confirmed the findings with measurements in which the surface potential was varied with a given metal [34]. The results for three different proteins, using combinations of gold, palladium and platinum electrodes, are shown in Figure 6. All three proteins studied show a resonance peak at about +300 mV on the normal hydrogen electrode scale (0 V on the NHE scale is −4.6 eV with respect to the vacuum [35], so +300 mV NHE is about −4.9 eV with respect to the vacuum). The existence of a resonance peak demonstrates that electrons are being injected into allowed states of the protein, so that transport is clearly not by tunneling (the data are fitted by Lorentzians based on the simplest model of resonant injection [34]). The energy of the stably ionized states of the component amino acids is given by their oxidation potential, and the most readily oxidized residues are tyrosine and tryptophan, for which the redox potentials are close to +1 V NHE [36,37]. This is 0.7 V removed from the observed transmission peaks.

## 7. Conformation and Conductance

What evolutionary driving force makes the interior of proteins better conductors than designed molecular wires? That question cannot be answered at present. However, if there is a functional role for long range conductance, then it is reasonable to suppose that conductance should change with changes in protein conformation. It was shown that the binding of biotin to streptavidin altered its conductance substantially [26]. In the case of multivalent proteins like streptavidin, with four available binding sites, it is possible to use two sites for electrical connections, leaving two sites open for the sensing of binding events. Biotin binds streptavidin very strongly, so the differences were manifest in stable bound and unbound molecules that did not change over the time course of an experiment. Our real interest lies in following enzyme activity in real-time. Given the lack of noise on the IV curves below 100 mV (Figure 3b), we reasoned that it should be possible to record the “noise” generated as an enzyme carries out its function, by biasing the enzyme with a voltage below 100 mV. However, in the case of an enzyme, with a single active site, it is necessary to engineer electrical contacts into the protein at two sites that do not interfere with its function. We chose the DNA polymerase Φ29, because of its potential utility as a single-molecule DNA sequencing device [38]. This enzyme, bound with template DNA (to be copied) a primer (defining the starting point in the template 3’ to 5’ direction at the double-strand-single strand junction) and necessary magnesium ion, is in the “open” conformation, as shown in Figure 7b. Once a matching nucleotide triphosphate (dNTP) is captured from solution, the enzyme snaps shut, going into the “closed” conformation, as shown in Figure 7c. It remains closed until the hydrolysis of the triphosphate chain, incorporation of the nucleotide monophosphate into the extending polymer, and translocation to the next incorporation site are complete. It then snaps open again in readiness for the next matching dNTP to arrive. In this enzyme, the N terminus is available for electrical connection, but the C terminus is close to the catalytic site. We identified a region of the exonuclease domain that did not move significantly over the open to closed transition, and incorporated a short peptide sequence there called an Avitag. The Avitag is biotinylated by an enzyme (BirA). An Avitag was also incorporated at the N terminus. The complete circuit (Figure 7a) was built by coating electrodes with a thiolated biotin (B), capturing streptavidin molecules (SA) and then bridging the gap with doubly biotinylated Φ29. This approach was adopted to keep the Φ29 well clear of the electrodes. The conductance of the complete circuit was only slightly less than that of a single streptavidin molecule, a consequence of the fact that the electrode contact resistance limits the overall conductance of these bioelectronic circuits, so that adding additional protein to the path lowers conductance only slightly. A distribution of conductances measured for the open state is shown in Figure 7d. The peak for the desired doubly connected geometry is located at 5.6 nS. The polymerase can be trapped in the closed configuration using non-hydrolyzable dNTPs, and the resulting distribution shows that the highest peak moves to 15 nS (Figure 7e). This peak is broadened and shifted to an intermediate conductance when regular dNTPs are used, in which case the polymerase is switching rapidly between open and closed conformations. A trace of current vs. time (at a fixed bias of 50 mV) shows large fluctuations (Figure 7f). These are only observed when all the ingredients for polymerization are present [38]. It is difficult to analyze these signals on a STM, because the junction is not very stable. However, incorporating the system into a solid-state tunnel junction device allows for data collection over long period of time, as well as for the recording of current changes in response to chemical changes. We expect that we will be able to measure the various rate constants for steps in the polymerization process quite accurately.

## 8. Mechanisms

It is known that long-range electron transport is possible via the hopping of radical cations in photoexcited proteins [39]. However, as discussed above, injected electrons from metal contacts are far removed in energy from the potential required to ionize the aromatic amino acids. What are the states that mediate transport in this case? Matyushov [40] has shown that the electrostatic fluctuations that localize charge (λ in Figure 1c) occur over much broader (and slower) time-scales than is the case in low molecular weight solvents like water. Thus, if the hopping rate (determined by τ) is fast enough, the transitions are non-ergodic and the full electrostatic reorganization is not sampled at each site. Such a mechanism was first proposed as an explanation of long-range transport in redox proteins by Kuznetsov and Ulstrup [41]. The outer hydrophilic regions of the protein are heavily hydrated, so more like the isolated hydrated peptides that are such good insulators (Figure 5), a fact that might account for the significant charge injection barrier at the surface of the protein. Since the energy of the fully relaxed states are known for the aromatic amino acids (~1V NHE oxidation potentials), and since λ in water is on the order of 1eV, it is quite possible that the resonant states at 300 mV NHE are unrelaxed molecular states of the aromatic amino acids. The interiors of the proteins we have examined all contain a number of tyrosines and tryptophans. However, their density is not large; in streptavidin, 12 out of 123 amino acids are tyrosines or tryptophans (10%) in polymerase Φ29, it is 49 out of 575 (9%), and in one arm of the IgE, it is 8 out of 230 (3.5%). While the presence of these residues might account for the energy of the observed transmission resonance (though there is no correlation between the absolute numbers of Y and W residues and the peak potentials), it is hard to see how a few dispersed aromatic residues could outperform the closely stacked aromatic moieties found in synthetic molecular wires. Yet, at least in the exponential model, the conductance per unit length is ~exp(10)= 20,000 times that of a molecular wire, and ~exp(45) = 10^19^ times that of a hydrated peptide chain. In the case of break-junction studies of peptides, this discrepancy might be trivially explained by the fact that the peptides examined to date [29,31,32,33] do not contain tryptophans or tyrosines (but if they did, the residues would be in contact with water, and therefore subject to rapid electrostatic relaxation). However, this trivial explanation would not account for the high conductivities of proteins relative to molecular wires, for distances in excess of 7 nm (Figure 2 and Figure 5). Is there something special about the structure of a folded protein? An examination of the distribution of calculated energy-level spacings in a number of materials [42] found that functional proteins followed neither the distribution characteristic of insulators, nor that characteristic of metals, but rather an intermediate distribution characteristic of a quantum-critical material, i.e., at a point right at the metal insulator transition as a function of the degree of disorder of the system. It is hard to see how this quantum mechanical description might apply to a hydrated protein at 300 K, but perhaps there is some robust feature of the folded geometry that supports an unusual electronic state, even in the presence of fluctuations.

## 9. Conclusions

It is clear that many non-electron transfer proteins are excellent conductors, if connected to electrodes by strong chemical bonds, preferably using ligands that can inject charge into the hydrophobic interior of the protein. Charge injection from the Fermi energy of metal electrodes is resonant with internal molecular states of the proteins. The energy of these molecular states is almost the same in different proteins, implying a common origin in certain amino acid residues. If these states are associated with tyrosines and tryptophans (as seems most likely), then they reflect the loss of the (dissipative) electrostatic reorganization energy of about 0.7 eV. Turning-off the loss of energy to the environment through reorganization might account for the long-range propagation of electrons. The conductance per unit length is strikingly higher than that of a typical molecular wire (for distances >7 nm), and the measured conductivity of proteins containing no electroactive components is only marginally less than that of a protein filament containing closely stacked hemes [3]. The mechanism of this remarkable conductance is not at all clear. If it is associated with some special feature of the geometry of folded proteins [42], that begs the question of just what the evolutionary driving force could be for the selection of such a special geometry. These questions will drive future research. Additional data will come from measurements of the temperature dependence of the conductance of single molecules and measurements on long aggregates that are strongly coupled. However, the utility of the phenomenon is clear. As illustrated by the circuit we have assembled to read out polymerase fluctuations, proteins make robust electronic components. At the nano-scale, they have the ability to self-assemble, as illustrated in Figure 7a. At the atomic scale, every atom can be positioned precisely by means of recombinant engineering. Given their remarkable electronic properties, and the fact that a wide range of enzymatic chemical functions can be transduced directly into an electrical signal, protein-based bioelectronics will surely find widespread application.

## Figures and Tables

**Figure 1 life-10-00072-f001:**
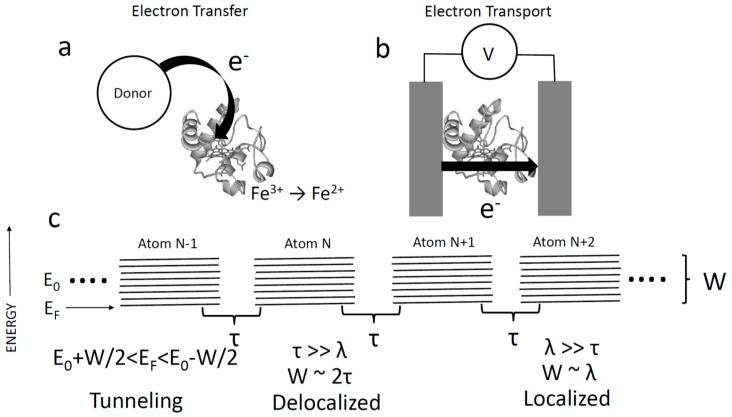
(**a**) Electron transfer is the process whereby an electron is transferred to/from a site on a protein that becomes ionized (reduced or oxidized). (**b**) Electron transport is the process whereby electrons pass from one electrode (or donor) to another (or acceptor) via a protein without residing on the protein. (**c**) Tight binding model of electron transport consists of an infinite chain of atoms, each of orbital energy $E_{0}$ with an interaction energy between neighbors, τ. The ‘ladder’ of states represents the spread of atomic energies owing to interactions. In this simple model, states are delocalized and energies in the solid occupy an energy range $W=E_{0}\pm \tau$. In a molecular solid, the soft medium distorts to screen a localized charge, leading to a spread in electron energies of λ, the reorganization energy. When λ≫τ the bandwidth becomes dominated by λ, and electrons are localized. Both limits assume that the electron is injected into allowed states, i.e., $E_{F}$ lies in the range of $E_{0}\pm W/2$. If this is not the case, transport is by tunneling and restricted to distances of a few Angstroms.

**Figure 2 life-10-00072-f002:**
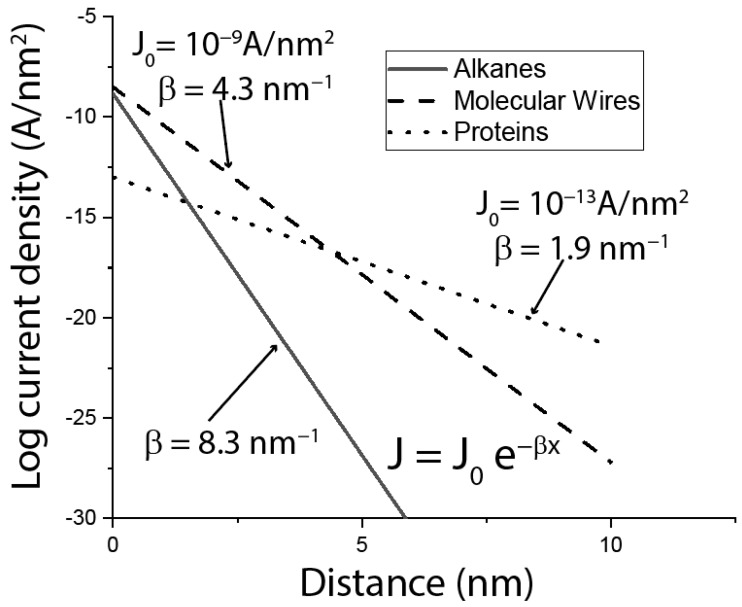
Summary of experiments that measured the current density through layers of alkanes (solid line), molecular wires (molecules containing a high density of π oritals—dashed line) and proteins (dotted line). The lines are fits to the data points in Figure 2 of Amdursky et al. [11].

**Figure 3 life-10-00072-f003:**
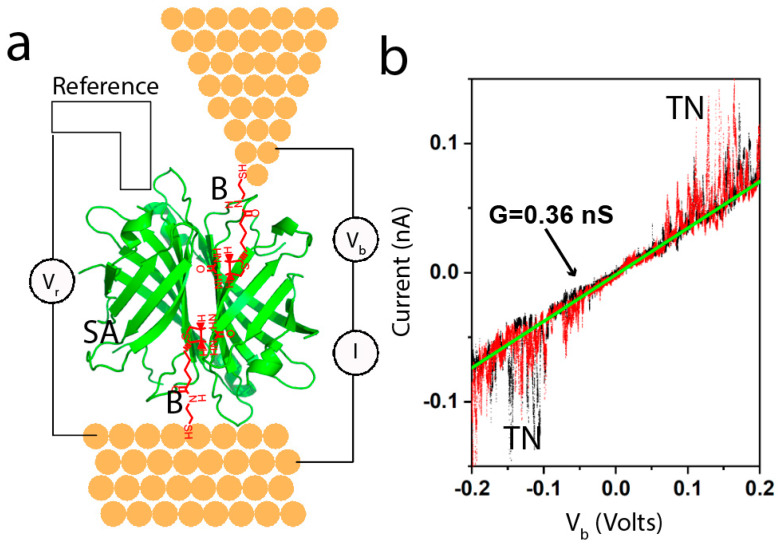
(**a**) Example of the experimental set-up for electrochemical STM studies of single-molecule conductance. The electrodes are functionalized with thiolated biotin molecules (B—red on the diagram) shown here trapping a streptavidin protein (SA—green). The electrodes are held under potential control at a bias V_r_ with respect to a reference electrode submerged in the electrolyte solution. (**b**) Example of a current-voltage curve obtained from a trapped protein by sweeping V and measuring I. The black data points are from the sweep up and the red data points are from the sweep down. The curve is linear, and its slope yields the conductance for the particular contact geometry in this case. These curves are acquired as a fixed gap traps the target molecule, largely eliminating the stresses and strains associated with break-junction measurements. Many such curves are used to compile distributions of single-molecule conductance. Here, TN stands for telegraph noise. Above ±100 mV, the protein contacts generate random noise that fluctuates between two levels of current at a fixed bias.

**Figure 4 life-10-00072-f004:**
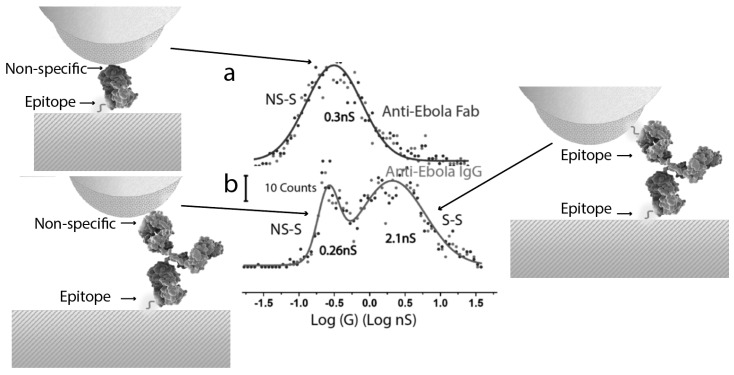
Conductance distributions for (**a**) an antibody Fab fragment and (**b**) the full antibody in contact with electrodes functionalized with a small peptide that is an epitope for the antibody. The vertical axes are shifted for clarity, the scale bar showing 10 counts. The high conductance peak in (**b**) comes from two specific contacts where the antibody is bound to epitopes on each electrode (illustrated on the right). The smaller conductance peak comes from an antibody or Fab fragment that is only bound specifically at one of the two electrodes (as illustrated on the left).

**Figure 5 life-10-00072-f005:**
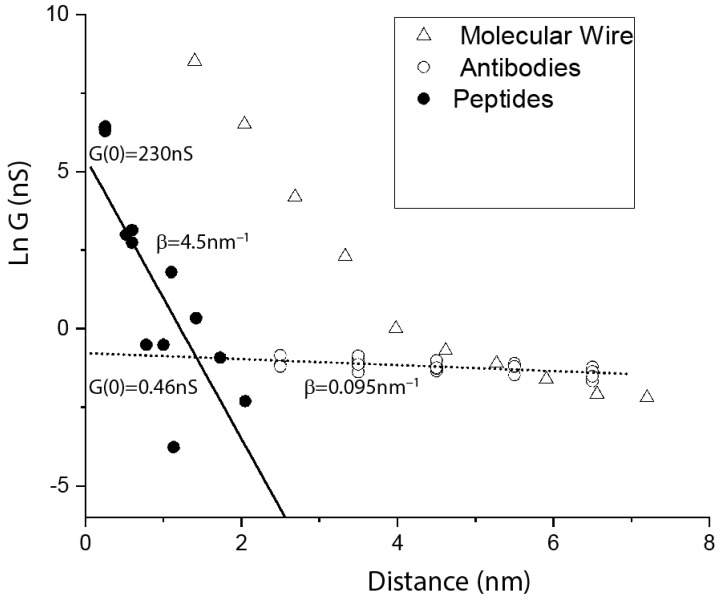
Plot of the natural logarithm of the conductance of single molecules versus the length of the molecules. Data for small, hydrated peptides of various lengths (filled circles) come from break-junction experiments [29,30,31,32,33]. Data for intact proteins (open circles) come from experiments in which one specific contact and a second nonspecific contact were made to antibodies or Fab fragments at various electrode gap sizes [27]. Data for oligophenylineimide molecular wires of various lengths [28] are shown as triangles. In the tunneling regime, the molecular wire data are fitted by β= 3 nm^−1^. In the hopping regime, they yield β= 0.9 nm^−1^. This decay constant is an order of magnitude larger than that for the proteins (β= 0.095 nm^−1^). This single-molecule data set qualitatively follows the patterns observed in macroscopic devices (Figure 2). Proteins have by far the smallest electronic decay rate, but much higher contact resistance than other molecules.

**Figure 6 life-10-00072-f006:**
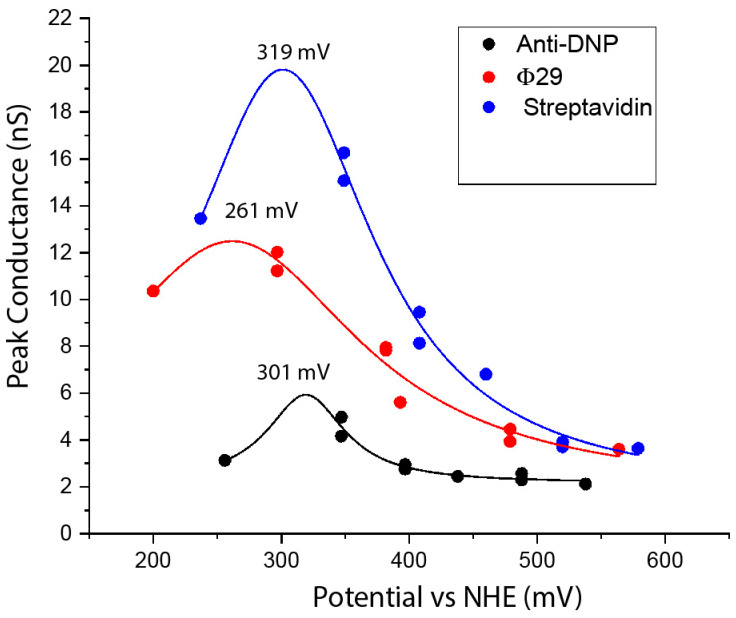
Conduction resonances in three proteins (see legend), showing peak conductance versus the electron injection potential, calculated from rest potential measurements [34]. The existence of a resonance demonstrates that electron injection from the Fermi energy of the metals is directly into a set of molecular states, and that the middle of the band of states is located at about 300 mV on the NHE scale. Solid lines are fits to Lorentzian functions with the peak values indicated.

**Figure 7 life-10-00072-f007:**
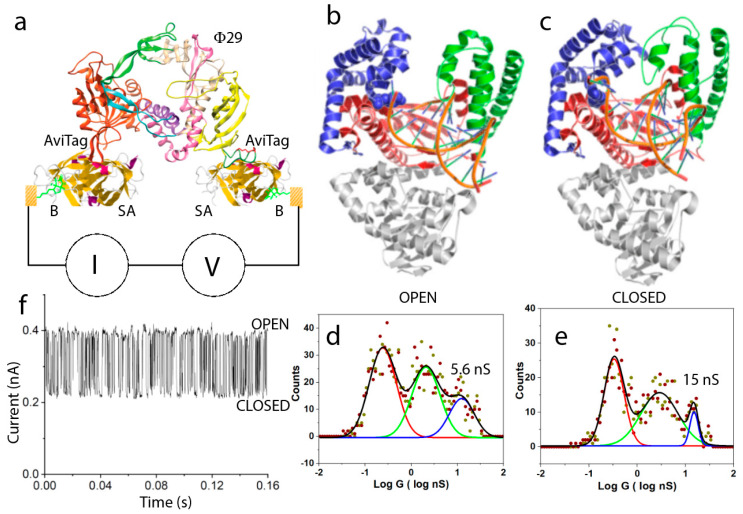
Conductance changes with functional motions of an enzyme. (**a**) Assembly of a bioelectronic circuit for monitoring the conductance of an active enzyme (Φ29 DNA polymerase). Biotinylated (B) electrodes capture streptavidin (SA), which in turn captures a doubly-biotinylayed polymease at regions where the biotinylated Avitag sequence has been inserted. (**b**) In the inactive state, the polymerase is in the open conformation. It retains this conformation on binding the DNA template and primer. (**c**) Once a nucleotide triphosphate that is complementary to the site on the template at the catalytic site of the polymerase is captured, the polymerase snaps into the closed conformation (**b**), where it remains until the nucleotide addition is complete. (**d**) Conductance distribution for open polymerases. (**e**) Conductance distribution for polymerases trapped in the closed conformation with the use of non-hydrolyzable dNTPs. When normal, hydrolyzable dNTPs are presented to the primed polymerase, it fluctuates rapidly between open and closed states as the template is copied, giving rise to rapid current fluctuations (**f**).

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
