# Peer review of "Ubiquitous Electron Transport in Non-Electron Transfer Proteins"

_life, 2020, doi:10.3390/life10050072_

Round 1

Reviewer 1 Report

 A timely, very interesting and much expected review of the new, emerging field of BioMolecular Electronics. Undoubtedly, this review is a must (and others to follow) as a compilation of latest advances towards a more in-depth understanding of the fascinating, yet puzzling, long-range electron transport in biology. This report will serve all members of this community to discuss experimental discrepancies from very different approaches and decide on fitting theories. My decision here is publishing the work as it is. In what follows, I intend to provide few suggestions to the author in my opinion might help enrich a bit the manuscript. These suggestions do not impose any condition over publication and they are totally up to the author to take them into consideration. It is my desire to see this review out as soon as possible.

  • The review focuses on ubiquitous electron transport in non-redox proteins. The reasons for why the authors decided to study non-redox protein are well-stated (a comment on this point is given below). However, section 2 refers to previous results on conductive proteins, most of them bearing indeed redox co-factors. This should be perhaps clarified. I would have perhaps expected a bit more recap on previous studies on redox proteins in this section (including the already mentioned Cahen's results), which constituted the kickoff of the actual field. Some of these works provide info on the non-redox conductivity by comparing hollow and apo protein systems, I think directly connected and very appealing to the current review topic.
  • In line 91 (same section 2), I found a bit raw the statement "proteins (? = 1.9 nm−1) outperform molecular wires (? = 4.3 nm−1)" without a proper explanation on the concept of "long-range" transport.
  • Figure 2 refers to decay current for "molecular wires" systems. I believe this should be defined as to what kind of chemistry is inferred there.
  • Section 3 (line 102) references an initial list of reviews [9-11]. I might be wrong but ref 11 is not one of them. I feel like the references to previous works on single-protein conductance (some dating back from 2011/2012) should be explicitly listed here, rather than just an indirect mention to a review, specially considering they are not that many and will provide the reader with the wide picture of the field.
  • Down in the same section (line 110), the statement "We were concerned that some (unknown) nanoscale mechanism might involve transport via rapid reduction and oxidation of the redox active sites" deserves I think further clarification. Most electron transfer proteins carry redox co-factors, which, in my opinion, might still be a key ingredient in the long-range mechanism, being the mentioned bacteria filaments a good example of this.
  • Section 8 beautifully discusses plausible mechanisms in light of current results. Line 299 states "resonant states at 300 mV NHE are unrelaxed molecular states of the aromatic amino acids". This connects directly to the formalism of Ulstrup and Kutnesov referred as 2-step sequential tunneling with partial relaxation. This formalism has been repeatedly used to understand fast "hopping" through redox-proteins and would be nice mention here as it connects to the nature of the mentioned resonances.
  • Finally, conclusions start off with the nice statement "proteins are excellent conductors, if connected to electrodes by strong chemical bonds". It would look very informative if put together with the various ways this has been done; namely thiols connecting directly to the protein structure, amide bonds, surface functionalization, etc.

Few typos spotted:

  • Figure 1 caption (line 42) says "cosnsists" should be "consists" ?
  • Figure 4 left bottom panel appears to have the arrows shifted downwards ?

Reviewer 2 Report

Comments are attached

Reviewer 3 Report

The author provides and interesting and thorough review of electron transport through proteins and how their structure can increase or decrease that rate. He then shows some interesting applications of this work measuring protein dynamics and related function in a DNA polymerase enzyme. Overall the work is well researched and presents many interesting avenues for further research and for others to build off of.

I had a hard time understanding line 203. Please correct that sentence.

Figure 4 was not direct. Please make the comparisons between 4a and 4b more clear. The y-axis change was not apparent at first viewing and I could not get through it easily.

Lines 190-95: I think this point would benefit greatly from a small discussion of the peptide sequences used. There is some discussion about the differences in contact resistance as a function of the types of amino acids (hydrophobic vs hydrophilic). A short discussion about the peptides used and how they could possibly fold and change the electrochemical properties would be very helpful here.

Reviewer 4 Report

This is a timely review regarding the important idea of electron transport in proteins. The author rightly challenges the idea concerning protein conductance in non-redox active proteins. Their observations lay down the foundation for new approaches to analyse protein function down to the single molecule level, including experimental setup. The work with the DNA polymerase is particularly impressive. 

My only point to address is lack of acknowledgment of related electrochemical STM studies on redox proteins, especially those where there is direct electrode bridging (e.g. DOIs 10.1021/jacs.7b06130, 10.1039/c2nr32131a).

Other than that, a timely and interesting review. 

Author Response

The two references cited have been added to the review.  Thank you.